# Optical Window to Polarity of Electrolyte Solutions

**DOI:** 10.3390/molecules28114360

**Published:** 2023-05-26

**Authors:** Omar O’Mari, Valentine I. Vullev

**Affiliations:** 1Department of Bioengineering, University of California, Riverside, CA 92521, USA; ooma001@ucr.edu; 2Department of Chemistry, University of California, Riverside, CA 92521, USA; 3Department of Biochemistry, University of California, Riverside, CA 92521, USA; 4Material Science and Engineering Program, University of California, Riverside, CA 92521, USA

**Keywords:** Stokes’ shift, dipole, absorption, fluorescence, refractometry

## Abstract

Medium polarity plays a crucial role in charge-transfer processes and electrochemistry. The added supporting electrolyte in electrochemical setups, essential for attaining the needed electrical conductivity, sets challenges for estimating medium polarity. Herein, we resort to Lippert–Mataga–Ooshika (LMO) formalism for estimating the Onsager polarity of electrolyte organic solutions pertinent to electrochemical analysis. An amine derivative of 1,8-naphthalimide proves to be an appropriate photoprobe for LMO analysis. An increase in electrolyte concentration enhances the polarity of the solutions. This effect becomes especially pronounced for low-polarity solvents. Adding 100 mM tetrabutylammonium hexafluorophosphate to chloroform results in solution polarity exceeding that of neat dichloromethane and 1,2-dichloroethane. Conversely, the observed polarity enhancement that emerges upon the same electrolyte addition to solvents such as acetonitrile and *N*,*N*-dimethylformamide is hardly as dramatic. Measured refractive indices provide a means for converting Onsager to Born polarity, which is essential for analyzing medium effects on electrochemical trends. This study demonstrates a robust optical means, encompassing steady-state spectroscopy and refractometry, for characterizing solution properties important for charge-transfer science and electrochemistry.

## 1. Introduction

### 1.1. Background

Medium polarity strongly affects the behavior of charged and dipolar species. The solvation in condensed media governs the dynamics of processes involving charge transfer (CT), ranging from the intramolecular redistribution of electron density to heterogeneous CT at electrode surfaces [1]. Polarity depends on the susceptibility of the medium to polarize under electric field, and it affects various properties of the solvated species, such as their solubility, reactivity, electrochemical potentials, and spectral transitions [2,3,4,5,6,7,8,9,10].

Born polarity, accounting for the solvation energy of charges and inversely proportional to the solvent dielectric constant, provides a good quantitative description of medium effects on electrochemical potentials, CT driving forces, and outer-sphere reorganization energy [11,12,13]. In fact, not only solvent polarity, but also the concentration of the supporting electrolyte (*C_el_*) significantly affects experimentally obtained electrochemical potentials [14,15]. Dissolved ionic species do modulate solution polarity. (1) Dissolving an electrolyte comprising kosmotropic ions in aqueous media tends to decrease its polarity [16,17]. (2) Chaotropic ions, on the other hand, can increase the polarity of solutions, especially those composed of low-polarity solvents [18,19,20,21,22,23]. 

Capacitance measurements offer a robust means for determining dielectric constants of non-conducting solvents and materials [24,25,26]. Adding electrolytes to the solvents for attaining the conductivity needed for electrochemical measurements, however, presents challenges. Since media polarity affects optical transition to and from highly polarized states, molecular photoprobes offer a convenient handle for examining the polarity of electrolyte solutions that solvated species experience. Such optical reporters account for the polarity of only the media that is immediately around the solvation cavity. Nevertheless, such estimates of “localized” polarity are considerably more valuable for molecular and nanometer-scale systems than the average bulk properties of the solutions.

As important as Born polarity is for characterizing medium effects on long-range CT and on reduction and oxidation processes, it fails when it comes to optical transitions involving changes in electric dipole moments. Conversely, the Onsager reaction field theory provides a good model for quantifying the solvation of dipoles [27]. Based on the Onsager model, the work of Wolfgang Lippert, Mitsuo Mataga, and Yasuteru Ooshika provides a means for estimating differences between the electric dipoles of ground and the emissive excited states [28,29,30,31].

The Lippert–Mataga–Ooshika (LMO) formalism is invaluable for extracting information about excited-state dipoles from absorption and emission spectra recorded for media with different polarities [32,33,34]. Conversely, resorting to coumarin 153 (C153), we recently demonstrated the utility of employing the LMO formalism in reverse for characterizing the polarity of electrolyte solutions widely used in analytical electrochemistry [35]. The inherently large dipole change upon photoexcitation makes chromophores with emissive CT states particularly attractive for LMO analysis, which justifies our choice for C153. Nevertheless, the excited-state behavior of C153 limits its utility for LMO analysis to solutions that are more polar than dichloromethane (CH_2_Cl_2_). This outcome warrants the search for different photoprobes that allow analyzing a broad range of medium polarity pertinent to electrolyte solutions important for electrochemistry, and, most of all, an improved understanding of what constitutes a good photoprobe for LMO analysis. 

Herein, we demonstrate the utility of 4-dimethylamino-*N*-phenyl-1,8-naphthalimide (ANI-Ph) for spectroscopic estimations of the Onsager polarity of electrolyte solutions in solvents ranging from relatively polar acetonitrile (CH_3_CN) to medium-polarity chloroform (CHCl_3_), where a commercially available electrolyte is soluble. The measured refractive indices of the electrolyte solutions allow us to extract their Born polarities from the LMO-determined Onsager ones, since the former have pertinent importance for electrochemical applications. Density-functional theory (DFT) reveals the electronic characteristics of the ground and excited states of ANI-Ph, which elucidates the chromophore features needed for LMO analysis. Therefore, with well-selected photoprobes, this optical method proves promising for determining the polarity of electrolyte solutions.

### 1.2. Born Polarity vs. Onsager Polarity: What’s the Difference?

Three sources of polarization govern medium polarity: (1) orientational polarization, **P_μ_**, reflecting on the ability of the solvent electric dipoles to orient along the applied field; (2) vibrational polarization, **P***_ν_*, depending on the field-induced redistribution of the positively charged solvent nuclei; and (3) electronic polarization, **P***_e_*, accounting for the field-induced shifts in the electron density of the solvent molecules [36]. Numerous scales for the quantification of medium polarity have emerged through the years that rely on bulk solvent characteristics such as static dielectric constant, *ε*, and index of refraction, *n* [2,3]. While *ε* accounts for medium **P_μ_**, **P***_ν_*, and **P***_e_*, *n* depends only on **P***_e_*. That is, when the applied electric field oscillates too quickly for the medium dipoles and nuclei to move and react to it, the polarity depends only on **P***_e_*; thus, *n*^2^ is referred to as optical or dynamic dielectric constant. 

Born solvation energy, Δ*G_B_*, represents the stabilization of a spherical charge with a radius, *r*, in a medium with dielectric constant, *ε* [11]:(1a)$${\Delta G}_{B}=\frac{{\left({zq}_{e}\right)}^{2}}{{8\pi \varepsilon }_{0}r}f_{B}\left(\varepsilon \right)$$
where *z* is the charge of the solvated species, *q_e_* is the elementary charge, *ε*_0_ is the vacuum permittivity, and *f_B_* is the Born polarity of the solvating media: (1b)$$f_{B}\left(\varepsilon \right)=\left(1-\frac{1}{\varepsilon }\right)$$

Accounting for Δ*G_B_* is essential for understanding the medium effects on the CT processes involved, for example, in electrochemistry and (photo)redox catalysis. For example, *f_B_* is a part of the Rehm–Weller equation for characterizing the thermodynamics of photoinduced CT [13,37], and the Pekar factor, *f_B_* is essential for estimating the outer sphere, or medium, CT reorganization energy [12,38]. 

When opposite charges are too close within the same solvation cavity forming a dipole, Δ*G_B_* cannot quantify their solvation energy, warranting the use of the Onsager model instead. It accounts for the mutual polarization of dipolar species and the surrounding solvent. The Onsager model includes electrostatic terms arising from the permanent dipole moment and the volume of the solvated species. It defines the Onsager polarity, *f_O_*, encompassing **P_μ_**, **P***_ν_*, and **P***_e_*, as:(2a)$$f_{O}\left(\varepsilon \right)=2\frac{\varepsilon -1}{2\varepsilon +1}$$
and *f_O_* accounting for **P***_e_* only is:(2b)$$f_{O}\left(n^{2}\right)=2\frac{n^{2}-1}{{2n}^{2}+1}$$

Thus, to extract the effects of the prominent polarizability arising from **P_μ_** when it is far greater than **P***_ν_*, i.e., |**P_μ_**| >> |**P***_ν_*|, the expression for the Onsager reaction field polarity becomes [29]:(2c)$$f_{O}\left(\varepsilon ,n^{2}\right)=f_{O}\left(\varepsilon \right)-f_{O}\left(n^{2}\right)=2\left(\frac{\varepsilon -1}{2\varepsilon +1}-\frac{n^{2}-1}{2n^{2}+1}\right)$$

The Onsager model assumes point–dipole approximation. That is, it treats the solute dipole as a point in the center of the spherical cavity. Nevertheless, this approximation still proves fairly applicable and extends to hydrogen-bond-forming solvents and non-polar media [29,39,40]. Moreover, the Onsager reaction field polarity relates to the Born polarity by solely considering the static dielectric constant:(3a)$$f_{B}\left(\varepsilon \right)=\frac{{3f}_{O}\left(\varepsilon \right)}{{2+f}_{O}\left(\varepsilon \right)}$$
(3b)$$f_{O}\left(\varepsilon \right)=\frac{{2f}_{B}\left(\varepsilon \right)}{3-f_{B}\left(\varepsilon \right)}$$

The Stokes’ shift refers to the energy difference between the maximum of the absorption and emission spectral bands originating from transitions between the ground and the same excited state of a chromophore. It represents the energy differences between Franck–Condon (FC) and relaxed electronic states. This relaxation often involves intramolecular charge redistribution. As a response to the thus-induced changes in the electric fields localized around a chromophore following optical transitions, **P_μ_** of the medium describes its reorganization around the solvation cavity. That is, as the orientation of the molecules of the solvating media does not change around the solute during light absorption and emission, it only reaches equilibrium during the relaxation of the FC states. Hence, the extent of stabilization of the relaxed states depends on the dipoles of the solvent molecules and their ability to reorient under an applied electric field. This feature allows the LMO formalism to relate the Stokes’ shifts (Δℰ), obtained from absorption and emission spectra, to the Onsager polarity of the media (Equation (2c)):(4)$$\Delta ℰ=\frac{{\left(\Delta \mu \right)}^{2}}{{4\pi \varepsilon }_{0}r^{3}}f_{O}\left({\varepsilon ,n}^{2}\right)+\Delta {ℰ}_{0}$$
where Δ*μ* is the difference between the magnitudes of the dipoles of the emissive excited state (**μ***) and the ground state (**μ**_0_), i.e., Δ*μ* = |**μ***| − |**μ**_0_|; and Δℰ_0_ is the Stokes’ shift for non-polar media with equal static and dynamic dielectric constants, i.e., *ε* = *n*^2^.

The LMO formalism relies on two types of solvent effects: (1) stabilization of the FC ground state after the radiative deactivation, i.e., S_0_^(FC)^→S_0_, and (2) stabilization of the FC excited state to its emissive structure after the photoexcitation. Therefore, it is important to combine LMO with other experimental and theoretical techniques in order to extract meaningful information about the electronic properties of a molecule from the dependence of its Stokes’ shift on solvent polarity.

## 2. Results

### 2.1. Selection of a Photoprobe

The LMO analysis does not account for changes in dipole orientation. That is, a principal assumption is that **μ*** and **μ**_0_ are parallel and point in the same direction, i.e., Δ*μ* = |Δ**μ**|, where |Δ**μ**| is the magnitude of the vector difference between the two dipole moments. That is, |Δ**μ**| = |**μ*** − **μ**_0_| or |Δ**μ**|^2^ = |**μ***|^2^ + |**μ**_0_|^2^ − 2 |**μ***| |**μ**_0_| cos(*α*), where *α* is the angle between the ground and excited-state dipoles. For chromophores that show good linearity of Δℰ vs. *f_O_*(*ε*, *n*^2^), it is important for *α* to be as small as possible, i.e., *α*→0, or ideally *α* = 0. Estimating the angle between μ* and μ_0_ to characterize such chromophores requires alternative methods, such as quantum-mechanical calculations. 

Another assumption of the LMO analysis is that the dipole change, Δ*μ*, is invariant to the medium polarity, i.e., Δ*μ* is introduced as a constant to the slope of the linear dependence of Δℰ vs. *f_O_*(*ε*, *n*^2^) (Equation (4)). Considering the reaction field inside the solvation cavity warrants an increase in the solute dipole with an increase in medium polarity. Because the ground and excited-state dipole are different, such a polarity-induced increase will also be different and preclude any invariance of Δ*μ* to *f_O_*(*ε*, *n*^2^). Therefore, it is common to observe the nonlinear behavior of Δℰ vs. *f_O_*(*ε*, *n*^2^), especially with the lowering of the solvent polarity [33,41]. Accounting for these considerations makes it crucial to define the LMO “working range” of a photoprobe where the polarity-induced change of Δ*μ* is minimal and Δℰ vs. *f_O_*(*ε*, *n*^2^) shows linearity within the experimental uncertainty. 

Selecting photoprobes that manifest large Δ*μ* values enhances the sensitivity of the LMO analysis since it increases the slope of the linear dependence of Δℰ vs. *f_O_*(*ε*, *n*^2^) (Equation (4)). Chromophores with emissive CT states present an excellent choice; therefore, voltage-sensitive dyes appear quite attractive for such applications. Nevertheless, a decrease in the medium polarity may destabilize the CT state and elevate its energy level above that of a locally excited (LE) state. Such changes in the nature of the emissive states usually result in alterations in the shapes of the spectral bands that prove detrimental for the LMO analysis because of changes in the orientation and the magnitude of the excited-state dipoles, **μ***.

Therefore, we focus on ANI-Ph, which has a fluorescent excited state with a moderate CT character. Along with similar naphthalimide derivatives, ANI-Ph demonstrates remarkable features that make its photophysical behavior highly sensitive to the solvating environment [42,43,44,45]. As a “push–pull” conjugate comprising an electron-donating amine and an electron-withdrawing imide attached to the naphthalene rings, ANI-Ph undergoes intramolecular CT in its excited state. As a result, the emissive excited state of ANI-Ph possesses a large dipole moment in, making its fluorescence pronouncedly solvatochromic. 

The natural transition orbitals (NTOs) for photoexcitation and radiative deactivation are delocalized predominantly over the three naphthalimide rings, with some extension over the dimethylamine (Figure 1). Solvent polarity has a negligible effect on the NTOs, not only for acetonitrile, but also for the least polar we selected for this study—i.e., chloroform. Moreover, the ground- and excited-state dipoles are practically parallel, with a ~2.7° angle between them that appears invariant to solvent polarity (Table 1).

The difference between the magnitudes of the excited- and ground-state dipoles is less than 4 D. Nevertheless, Δμ increases by only 10% upon transition from chloroform to the relatively polar N,N-dimethylformamide, DMF (Table 1). Despite the small Δμ, its minimal solvent dependence, along with the almost parallel orientation of **μ*** and **μ**_0_, makes ANI-Ph an attractive candidate for analyzing medium polarity using the MLO formalism.

### 2.2. Onsager Polarity of Electrolyte Solutions

In the presence of electrolytes in the solvating media, **P_μ_** encompasses the ability of the cations and anions to move and reorganize around the solvation cavity along with the reorientation of the solvent dipoles. Therefore, LMO analysis can provide information about the Onsager polarity of electrolyte solutions in the vicinity of the solvation cavity of the photoprobe, i.e., ANI-Ph. 

In pertinence to organic analytical electrochemistry, we focus on solutions of tetrabutylammonium hexafluorophosphate (N(*n*-C_4_H_9_)_4_PF_6_) as a broadly used supporting electrolyte in six aprotic solvents with different polarities: CHCl_3_, CH_2_Cl_2_, 1,2-dichloroethane ((CH_2_Cl)_2_), benzonitrile (C_6_H_5_CN), DMF, and CH_3_CN (Table 1). We vary the electrolyte concentration, *C_el_*, from 25 to 200 mM. 

Absorption and emission spectra of ANI-Ph in the neat solvents, i.e., *C_el_* = 0, provide a practical means for examining the performance of this dye as an LMO photoprobe and calibrate the linear dependence of its Δℰ on *f_O_*(*ε*, *n*^2^). Prior to extracting Stokes’ shifts for LMO analysis, it is key to transfer the measured absorption, *A*(*λ*), and fluorescence, *F*(*λ*), spectra from wavelength to energy scales by implementing transition dipole moment (TDM) corrections: i.e., *A*(ℰ) = *A*(*λ*)ℰ^–1^ and *F*(ℰ) = *F*(*λ*)ℰ^–5^ [46]. Such transformations change not only the relative spacing between the data points along the abscissa axes, but also the shapes of the spectra. 

Solvent polarity mainly affects the fluorescence of ANI-Ph, while inducing only slight variations in its absorption (Figure 2a–c). This finding suggests that the solvating media have similar effects on the ground, S_0_, and the Franck–Condon (FC) excited states, S_1_^(FC)^, in order to minimally affect the energy of the S_0_→S_1_^(FC)^ transition depicted by the absorption spectra. In contrast, polar solvents stabilize the emissive excited state, S_1_, more than they do the FC ground state, S_0_^(FC)^, as the solvatochromism of the fluorescence spectra, S_1_→S_0_^(FC)^, indicates (Figure 2c). 

While the polarity dependence of the absorption and the fluorescence maxima, ℰ(*A*_max_) and ℰ(*F*_max_), respectively, tend to deviate slightly from linearity (Figure 2c), the Stokes’ shifts of ANI-Ph manifest a strong linear correlation with the Onsager polarity of the neat solvents, i.e., *R*^2^ = 0.99 (Figure 2d). The slope from the linear fit allows extracting the ratio between Δ*μ*^2^ and *r*^3^, i.e., Δ*μ*^2^ *r*^–3^ = 0.027 e^2^Å^–1^ (Figure 2d). Considering the DFT calculated Δ*μ* is about 3.5 D = 0.73 e Å (Table 1), the effective radius, *r*, of ANI-Ph amounts to 2.7 Å. This value of *r* is twice smaller than the DFT estimation of the ANI-Ph radius, i.e., *a*_0_ = 5.4 Å. This discrepancy is consistent with the deficiency in the spherical approximation for the solvation cavity that the theory implements [15,36].

The effects of electrolyte concentration, *C_el_*, on the optical properties of ANI-Ph show the same trends as the effects of solvent polarity. Increasing *C_el_* from 0 to 200 mM causes bathochromic shifts that are larger for the fluorescence than the absorption (Figure 3a,b). This difference between the effects on the absorption and the emission leads to an increase in the ANI-Ph Stokes’ shift with an increase in *C_el_*. The slope and intercept from the linear fit of Δℰ vs. *f_O_*(*ε*, *n*^2^) for neat solvents (Figure 2d) allows us to estimate the Onsager polarity for the different electrolyte organic solutions using Equation (4) (Figure 3c,d). Increasing the concentration of the supporting electrolyte enhances the medium polarity that the dissolved molecular fluorophore experiences (Figure 3d). Lowering the solvent polarity enhances these *C_el_* effects (Figure 3). 

### 2.3. Born Polarity of Electrolyte Solutions

The Onsager reaction field, at equilibrium and optimally, follows the direction of the dipole. The different components of the medium polarization, however, do not exhibit the same dynamics when responding to the changes in the orientation and magnitude of solvated dipoles. The electronic polarization, **P***_e_*, as related to the medium polarizability and the dynamic dielectric constant, *n*^2^, almost “instantaneously” follows the dipole changes that optical transitions produce. The response of the vibrational and orientational polarizations, **P***_ν_* and **P_μ_**, of the solvating media, however, is in the picosecond time domain and even slower, which is comparable with the relaxation of FC to optimal geometry states. The LMO formalism, accounting for the difference between dipoles of relaxed excited and ground states, therefore encompasses only the “slow” **P***_ν_* and **P_μ_** components of the Onsager polarity of the solvating medium, i.e., *f_O_*(*ε*, *n*^2^) (Equation (2c)). 

To account for polarity effects on the solvation of charged species, which is important for CT and electrochemistry, it is crucial to include all components of medium polarization. To complement the results for *f_O_*(*ε*, *n*^2^) from the LMO studies, we resort to concurrent measurements of the refractive indices of the electrolyte solutions (see Appendix A). The indices of refraction allow estimating the dynamic dielectric component of the Onsager function, *f_O_*(*n*^2^), which accounts solely for **P***_e_* (Equation (2c)).

The added electrolyte only slightly perturbates the refractive indices and *f_O_*(*n*^2^) of the neat solvents (Figure 4a,b). Therefore, the electronic polarization and polarizability of the comprising ions, N^+^(*n*-C_4_H_9_)_4_ and PF_6_^–^, are quite similar to those of the used organic solvents. Following the fine electrolyte dependence of *n* reveals that for some solvents, such as CH_3_CN, increasing *C_el_* slightly enhances *n*^2^ and *f_O_*(*n*^2^). It indicates that the solvated electrolyte ions are more polarizable than these particular solvents. On the other hand, an increase in *C_el_* in other solvents, such as C_6_H_5_CN, slightly decreases the *n*^2^ and *f_O_*(*n*^2^) of the solutions, which is indicative of solvents having larger polarizability than the electrolyte ions. 

The sum of *f_O_*(*ε*, *n*^2^) and *f_O_*(*n*^2^) yields *f_O_*(*ε*) (Equation (2c), Figure 4c), and *f_O_*(*ε*) relates to *f_B_*(*ε*) (Equation (3a), Figure 4d). Affecting the reduction potentials of solvated species, quantifying the Born polarity, *f_B_*(*ε*), of electrolyte solutions is key for electrochemical and CT analysis [13]. While this study focuses on liquid solutions, the considerations from its outcomes are extendable to solid electrolytes with pertinence to energy science and engineering [47,48,49,50,51]. 

## 3. Discussion

Increasing the electrolyte concentration in organic solvents appears to enhance the solution polarity, accounting for the orientational and vibrational polarization, i.e., *f_O_*(*ε*), *f_O_*(*ε*, *n*^2^), and *f_B_*(*ε*), that depend on the static dielectric constant. Previous reports show that adding salts to aqueous media decreases the static dielectric constant [16,17]. Conversely, adding electrolytes to organic solutions tends to increase their polarity [18,19,20,21,22,23]. These opposing effects originate from the nature of interaction between the electrolyte ions and the solvent molecules. 

Salts comprising kosmotropic ions enforce the three-dimensional structure of solvating media, such as water, and impede the rotation of the solvent molecules and the orientation of their dipoles along permeating electric fields. An increase in the charge density of ions exerts similar suppression of the free rotation of dipolar solvent molecules. Decreasing the size or increasing the charge of ions enhances the strength of the electric field around them. It induces the electrofreezing of dipolar solvent molecules around the solvation cavity of ions with high charge density [52]. 

When specific interactions between electrolyte ions and solvent molecules are lacking, permeating electric fields reorganize the charged species. It enhances the orientational polarization and the static dielectric constant of the solutions without compromising the contributions from the solvent molecules. Therefore, adding an electrolyte comprising ions with small charge density, including chaotropes, increases the static dielectric constant of solutions, especially when the solvent molecules have small electric dipoles [35]. As a supporting electrolyte, N(*n*-C_4_H_9_)_4_PF_6_ comprises singly charged ions with relatively large van der Waals radii, i.e., small charge density, precluding strong electrostatic interactions with organic solvents broadly used in electrochemistry. It is consistent with the observed increase in *f_O_*(*ε*, *n*^2^) induced by increasing *C_el_* (Figure 3d). Furthermore, lowering solvent polarity makes this *C_el_*-induced trend especially pronounced (Figure 3d). 

The effect of the organic electrolyte, N(*n*-C_4_H_9_)_4_PF_6_, on the polarizability and *f_O_*(*n*^2^) of organic solvents is negligibly minute (Figure 4a,b). Therefore, the dependence of *f_O_*(*ε*) on *C_el_* shows trends quite similar to those of *f_O_*(*ε*, *n*^2^) vs. *C_el_* (Figure 3d and Figure 4c). This similarity extends to the *C_el_* dependence of *f_B_*(*ε*) (Figure 4d). Overall, *f_O_*(*ε*) and *f_B_*(*ε*) have similar dependence on *ε* (Figure 5a). Nevertheless, the relationship between *f_O_*(*ε*) and *f_B_*(*ε*) is by no means linear (Figure 5b). Thus, while optical methods provide a good means for estimating the Onsager polarity of the media, it is important to convert it to the Born polarity when analyzing the solvation of charged species (Equation (3)).

Static dielectric constants provide a facile, “convenient” means for estimating medium polarity. Nevertheless, *ε* is not linearly proportional to the solvation energy of dipoles and charges, as the inverse dependence of the Born polarity on *ε* illustrates (Equation (1b)). For example, changing the solvent from toluene to CH_2_Cl_2_ can have stronger impacts on spectroscopic and electrochemical properties than changing from CH_2_Cl_2_ to CH_3_CN. That is, transitions from toluene to CH_2_Cl_2_ and from CH_2_Cl_2_ to CH_3_CN enhance *ε* by about a factor of four. Nevertheless, the former increases *f_B_*(*ε*) by about 0.3, and the latter only by about 0.1. The effects of electrolyte concentration reflect these considerations of the nonlinear relationship between *ε* and the solvation energy. Lowering the solvent polarity enhances the effects of *C_el_*. Furthermore, the initial addition of electrolytes, up to about 100 mM, induces the largest increase in the solution polarity (Figure 3d and Figure 4c,d).

The nonlinear relationship between *f_B_*(*ε*) and *ε* points to an inherent limitation of the LMO method for evaluating medium polarity. The clearly measured increase in *f_B_*(*ε*) of chloroform from about 0.79 to 0.9 upon adding electrolyte (Figure 4d), for example, corresponds to doubling *ε* from about 4.8 to 10. Conversely, the same doubling of the static dielectric constant of acetonitrile solutions from 37.5 to 75 results in an increase in *f_B_*(*ε*) from about 0.97 to 0.99. Such a small increase in *f_B_*(*ε*) is challenging to unequivocally determine using the described method based on LMO analysis.

The precision of measuring the Stokes’ shifts and their susceptibility to changes in the medium polarity determines the sensitivity of this method for estimating *f_O_*(*ε*, *n*^2^) and, consequently, *f_B_*(*ε*). Sharp absorption and fluorescence peaks and a large difference between the dipoles of the ground and emissive excited states, Δ*μ* (Equation (4)), of the photoprobe improves the sensitivity of the LMO analysis. Nevertheless, these two requirements for a dye tend to be mutually exclusive. Tapping on fluorescence from CT excited states ensures large Δ*μ*. Introducing a CT character in an excited state, however, usually broadens its emission band. 

These considerations warrant caution when employing the LMO analysis to media with *f_B_*(*ε*) exceeding 0.95, i.e., with *ε* > 20. For solutions with low and moderate polarity, on the other hand, the LMO approach, combined with refractometry, proves truly promising for expedient quantification of their polarity. 

## 4. Conclusions

Optical methods, including steady-state spectroscopy and refractometry, provide a facile means for quantifying medium polarity. The selection of a fluorescent photoprobe with electronic properties that confine within the restrictions of the LMO analysis is key for the reliable extraction of information about solution polarity from absorption and emission spectra. Electrolytes increase the polarity of organic solutions. Lowering solvent polarity enhances this electrolyte effect. Adding electrolyte to chloroform increases its Onsager polarity, *f_O_*(*ε*, *n*^2^), by about 50%; to dichloromethane, by about 20%; and to acetonitrile, by about 2% (Figure 3d). The conversion between different solvation models, i.e., from Onsager to Born models, broadens the applicability of the results obtained from optical spectroscopy to electrochemical and charge-transfer analysis.

## Figures and Tables

**Figure 1 molecules-28-04360-f001:**
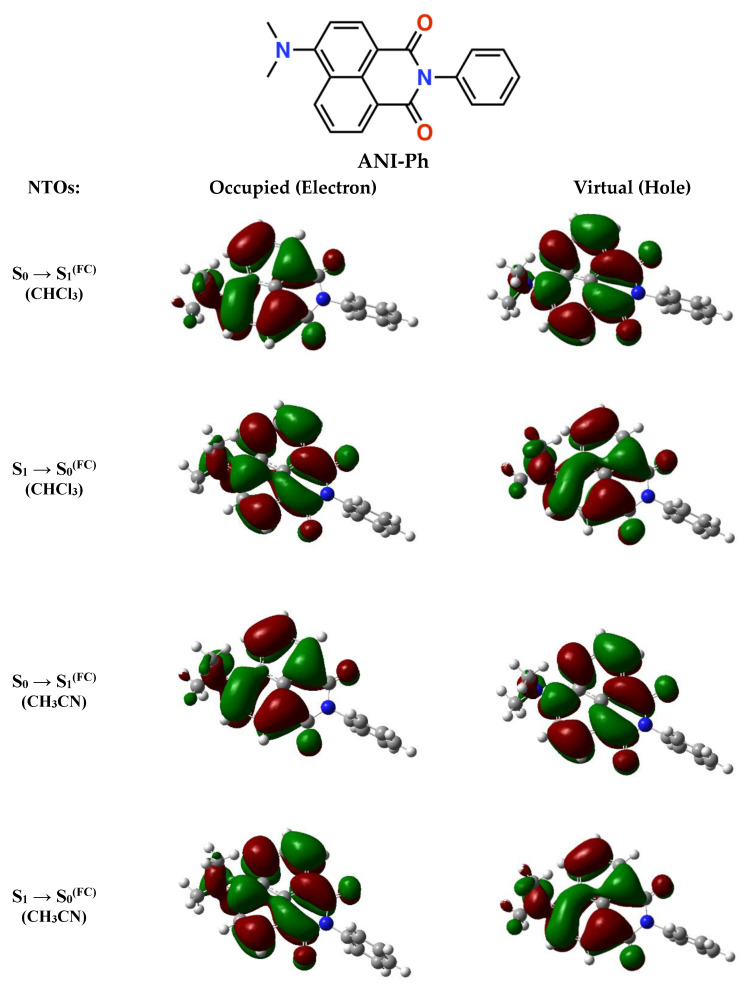
Structure of ANI-Ph, along with the natural transition orbitals (NTOs) for optical absorption, S_0_→S_1_^(FC)^, and emission, S_1_→S_0_^(FC)^, as obtained using the Hartree–Fock (HF) approach with a 6-31G(d,p) basis set and configuration interaction singles (CIS) method for the excited states (for details, see Appendix A). The NTOs were computed for CH_3_CN and CHCl_3_ as the integral equation formalism variant of the polarizable continuum model (IEFPCM) implements. The optimized electric dipoles of the ground and excited states differ by 3.30 D and 3.60 D for CHCl_3_ and CH_3_CN (Table 1).

**Figure 2 molecules-28-04360-f002:**
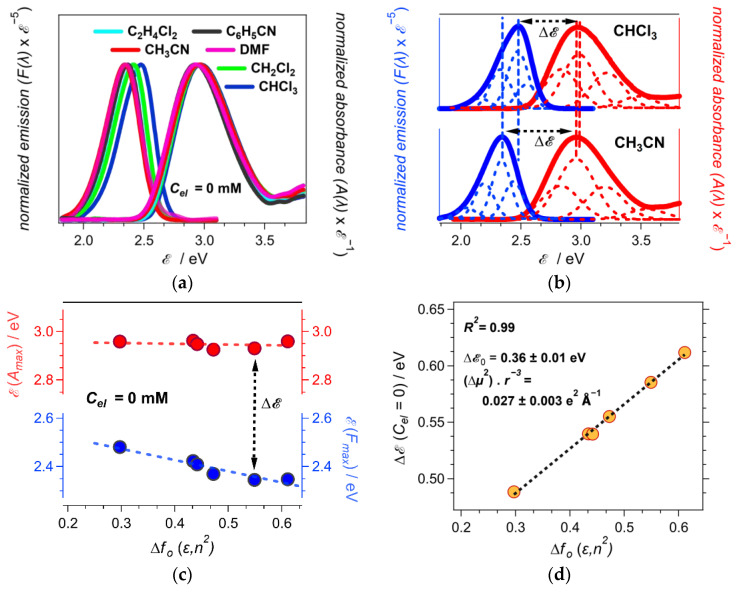
Steady-state optical spectra of ANI-Ph, along with the obtained Stokes’ shifts, Δℰ, for neat solvents. (**a**) Absorption and fluorescence spectra of ANI-Ph in different solvents corrected for the energy abscissa scale (*λ_ex_* = 421 nm). (**b**) Deconvolution of the absorption and fluorescence spectra of ANI-Ph for CHCl_3_ and CH_3_CN as a sum of Gaussians. The Gaussians with the predominantly larger amplitudes (the second from the spectral crossing point) are used for estimating the spectral maxima and the Stokes’ shifts. (**c**) Dependence of the absorption and fluorescence maxima of ANI-Ph on solvent polarity. (**d**) Linear fit of Δℰ vs. *f_O_*(*ε*, *n*^2^) for the neat solvents, i.e., *C_el_* = 0, along with the values of |Δ**μ**|^2^ *r*^–3^, extracted from the slope, and Δℰ_0_, extracted from the intercept.

**Figure 3 molecules-28-04360-f003:**
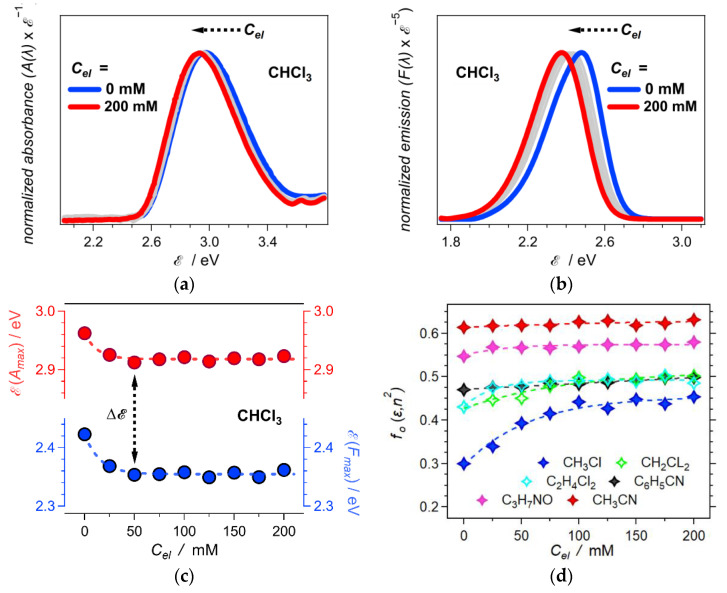
Effects of electrolyte concentration, *C_el_*, on solution polarity as determined from the LMO formalism (Equation (4)). (**a**) Normalized absorption spectra of ANI-Ph for CHCl_3_ in the presence of different concentrations of N(*n*-C_4_H_9_)_4_PF_6_ (for the optical spectra for the other solvents, see Appendix A). (**b**) Normalized fluorescence spectra of ANI-Ph for CHCl_3_ in the presence of different concentrations of N(*n*-C_4_H_9_)_4_PF_6_ (*λ_ex_* = 421 nm). (**c**) Dependence of the absorption and fluorescence maxima of ANI-Ph in CHCl_3_ on *C_el_*. (**d**) Dependence of the Onsager polarity, *f_O_*(*ε*, *n*^2^), of the electrolyte solutions on *C_el_* for the six different solvents (C_3_H_7_NO = DMF). The values of *f_O_*(*ε*, *n*^2^) for the different solvents and *C_el_* are obtained by introducing the Stokes’ shifts, ΔE, in Equation (4), along with the slope and the intercept that the linear analysis for neat solvents produces (Figure 2d).

**Figure 4 molecules-28-04360-f004:**
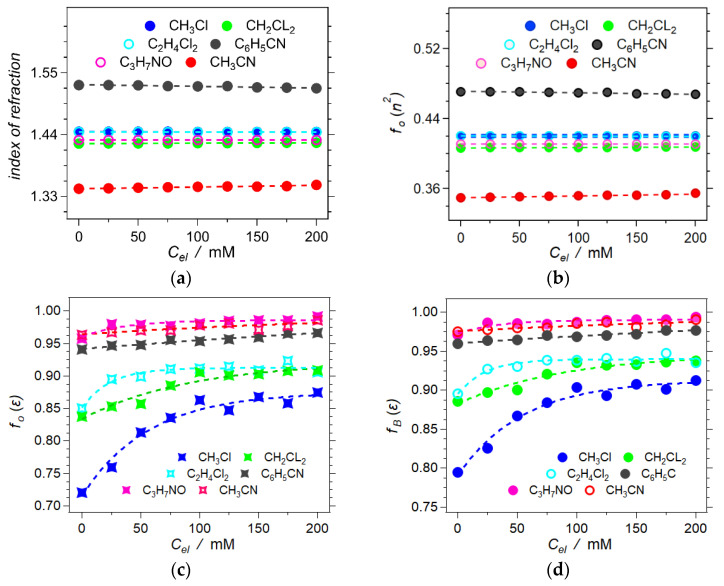
Resulting refractometry measurements and their Onsager polarity components (C_3_H_7_NO = DMF). (**a**) Experimental index of refraction for *C_el_* = 0 → 200 mM. (**b**) Onsager polarities of the dynamic dielectric, *n*^2^, obtained from the measured indices of refraction, *n*, Equation (2b). (**c**) Onsager polarities of the static dielectric, *ε*, obtained from LMO analysis, Equation (2c). (**d**) Dependence of the Born polarity, *f_B_*, on electrolyte concentration, *C_el_*, calculated from the relation Equation (3a).

**Figure 5 molecules-28-04360-f005:**
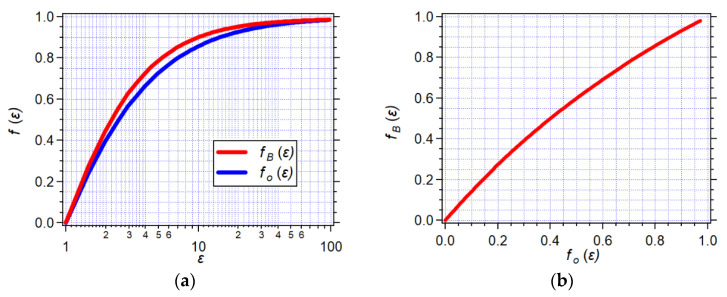
Relationship between the Born polarity, *f_B_*(*ε*), and the Onsager polarity, *f_O_*(*ε*), originating from the static dielectric properties of the media. (**a**) Dependence of the Born and Onsager polarities on the static dielectric constant. (**b**) Graphical representation of the relationship between *f_B_*(*ε*) and *f_O_*(*ε*).

**Table 1 molecules-28-04360-t001:** Permanent electric dipole moments of the ground and the excited state of ANI-Ph, obtained from a Hartree–Fock calculation at the 6-31g(d,p) basis set in the Polarizable Continuum Model (PCM) using the integral equation formalism variant (IEFPCM).

Solvent	*f_O_*(*ε*, *n*^2^) *^a^*	|μ_0_|/D *^b^*	|μ*|/D *^b^*	*α*/deg *^c^*	Δ*μ*/D *^d^*	|Δμ|/D *^e^*	|μ_0_|/D *^b^*
CHCl_3_	0.29	6.67	9.97	2.7	3.30	3.32	0.29
CH_2_Cl_2_	0.43	7.07	10.41	2.7	3.34	3.36	0.43
(CH_2_Cl)_2_	0.44	7.11	10.49	2.7	3.38	3.40	0.44
C_6_H_5_CN	0.47	7.29	10.87	2.7	3.58	3.60	0.47
DMF	0.55	7.33	10.95	2.7	3.62	3.64	0.55
CH_3_CN	0.61	7.33	10.93	2.7	3.60	3.62	0.61

*^a^* Onsager polarity determined from Equation (2c). *^b^* Magnitudes of the ground and excited-state dipoles, i.e., *μ* = |**μ**|. *^c^* Angle between **μ**_0_ and **μ***. *^d^* Difference between the magnitudes of **μ**_0_ and **μ***, i.e., Δ*μ* = |**μ***| − |**μ**_0_|. *^e^* Magnitudes of the vector differences between **μ**_0_ and **μ***, i.e., |Δ**μ**| = |**μ*** − **μ**_0_|, which is equivalent to |Δ**μ**|^2^ = |**μ***|^2^ + |**μ**_0_|^2^ − 2 |**μ***| |**μ**_0_| cos *α*. **μ**_0_ and **μ*** are practically colinear, with the angle between them less than 3°, |Δ**μ**| ≈ Δ*μ*.

## Data Availability

Data are all available upon Appendix A.

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
