# Peer review of "Optical Window to Polarity of Electrolyte Solutions"

_molecules, 2023, doi:10.3390/molecules28114360_

Round 1
Reviewer 1 Report
In this manuscript, Vullev et al. report an exciting and appealing methodology to measure the Born polarity, which is critical in analyzing medium effects on electrochemical trends. To do so, the authors mainly leverage the Stokes Shift and refractive index of a dye, which evidence some degree of charge transfer character. The results are convincing, and the manuscript is well-written. In this regard, the authors did a great job introducing the concepts needed to understand their approach to calculating Onsager polarity and Born polarity. It is with great enthusiasm that I recommend publication of this manuscript as it is.
Author Response
Comments:
In this manuscript, Vullev et al. report an exciting and appealing methodology to measure the Born polarity, which is critical in analyzing medium effects on electrochemical trends. To do so, the authors mainly leverage the Stokes Shift and refractive index of a dye, which evidence some degree of charge transfer character. The results are convincing, and the manuscript is well-written. In this regard, the authors did a great job introducing the concepts needed to understand their approach to calculating Onsager polarity and Born polarity.
Response:
The reviewer’s comments nicely capture the essence of this work. Thank you.
Comment:
It is with great enthusiasm that I recommend publication of this manuscript as it is.
Response:
Thank you so much for this recommendation.
Reviewer 2 Report
Ms. No.: Molecules-2393510
Title: Optical Window to Polarity of Electrolyte Solutions
A robust optical means was demonstrated to encompass steady-state spectroscopy and refractometry to characterize solution properties that are effective for charge-transfer science and electrochemistry. The proposed formula and models perfectly estimated the solution polarity using absorption and emission spectra. However, there are some issues that should be corrected before further consideration regarding following comments:
1. In present study Lippert–Mataga–Ooshika (LMO) formalism was used to estimate the Onsager polarity of electrolyte organic solutions pertinent to electrochemical analysis.
2. The introduction was written very well. The subject was described and the review of literature is adequate.
3. This manuscript presented an original research and provided useful strategy to evaluate the suitability of the polarity of the electrolyte organic solutions for electrochemical applications.
4. The applied optical method for selected photo probe was promising for determining the polarity of electrolyte solutions. So, this research introduced a new vision for determining the polarity of electrolyte solutions.
5. The methodology is correct and the whole procedure is acceptable in present format.
6. The figures were plotted well and their resolution are enough high.
7. The calculation were performed correctly and obtained results are acceptable. The presented data in conclusion part are consistent with the evidence and arguments presented in the manuscript and the main question was addressed properly. Meanwhile, it is recommended to add some quantitative data to this part.
8. All references are appropriate and cited in proper sections.
Author Response
Comments:
A robust optical means was demonstrated to encompass steady-state spectroscopy and refractometry to characterize solution properties that are effective for charge-transfer science and electrochemistry. The proposed formula and models perfectly estimated the solution polarity using absorption and emission spectra.
Response:
Thank you for the nice summary.
Comments:
However, there are some issues that should be corrected before further consideration regarding following comments:
- In present study Lippert–Mataga–Ooshika (LMO) formalism was used to estimate the Onsager polarity of electrolyte organic solutions pertinent to electrochemical analysis.
- The introduction was written very well. The subject was described and the review of literature is adequate.
- This manuscript presented an original research and provided useful strategy to evaluate the suitability of the polarity of the electrolyte organic solutions for electrochemical applications.
- The applied optical method for selected photo probe was promising for determining the polarity of electrolyte solutions. So, this research introduced a new vision for determining the polarity of electrolyte solutions.
- The methodology is correct and the whole procedure is acceptable in present format.
- The figures were plotted well and their resolution are enough high.
Response:
Thank you !!!
Comments:
- The calculation were performed correctly and obtained results are acceptable. The presented data in conclusion part are consistent with the evidence and arguments presented in the manuscript and the main question was addressed properly. Meanwhile, it is recommended to add some quantitative data to this part.
Response:
Thank you, again.
The reviewer also poses a reasonable request about quantitative data in the conclusions. To address it, we added three sentences on lines 391 to 394:
“Electrolytes increase the polarity of organic solutions. Lowering solvent polarity enhances this electrolyte effect. Adding electrolyte to chloroform increases its Onsager polarity, fO(ε, n2), by about 50%; to dichloromethane – by about 20%; and to acetonitrile – by about 2% (Figure 3d).”
This statement not only provides quantitative depiction of the found trends, but also illustrates a potential inherent limit of this method for polar media as we discussed in the section right before the conclusions.
Comment:
- All references are appropriate and cited in proper sections.
Response:
Thank you.
Reviewer 3 Report
In this manuscript, authors resort to LMO formatlism for estimating the onsager polarity of electrolyte organic solutions, which is essential for analyzing medium effects on electrochemical trends. In general, it is an interesting work and the manuscript is well organized. However, there are still some issues to be addressed. A moderate revision is suggested before its acceptance.
1. More solid data should be presented in abstract.
2. More background on the structure, properties and applications on electrolytes should be provided with supporting articles: Molecules 28 (5), 2042, 2023; Composites Communications 19 (239), 239-245, 2020; etc.
3. In Figure 1, the words electron and hole should be placed near to the corresponding image items, which will be better for understanding of this figure.
4. Why authors choose the listed solvents should be further clarified.
5. More comparison to previous work should be provided.
6. There are too many too old references, which is better to be deleted or replaced with recent articles to show the novelty of this work.
7. There are still some typos and grammar issues in the manuscript. Authors should carefully recheck the whole manuscript.
Author Response
Comments:
In this manuscript, authors resort to LMO formatlism for estimating the onsager polarity of electrolyte organic solutions, which is essential for analyzing medium effects on electrochemical trends. In general, it is an interesting work and the manuscript is well organized.
Response:
Thank you for the nice evaluation.
Comments:
However, there are still some issues to be addressed. A moderate revision is suggested before its acceptance.
- More solid data should be presented in abstract.
Response:
The reviewer brings up a valid point. On lines 15 to 19, we added a couple of sentences with results / data that support the other statements in the abstract.
Comment:
- More background on the structure, properties and applications on electrolytes should be provided with supporting articles: Molecules 28 (5), 2042, 2023; Composites Communications 19 (239), 239-245, 2020; etc.
Response:
In relevance to the effects of the electrolyte on solution polarity impotant for this study, we have already discussed the role of the chaotropic and kosmotropic nature of the comprising ions. Also, we briefly discussed the polarizability of the electrolyte ions in relevance to its effect on the refractive index of the solutions. The two references that the reviewer listed above are not directly pertinent to this study. Nevertheless, the reviewer brings an interesting point about the solid-statute electrolytes and their importance for energy storage. Therefore, we added a short statement about it on lines 312 and 313 with several references.
Comment:
- In Figure 1, the words electron and hole should be placed near to the corresponding image items, which will be better for understanding of this figure.
Response:
Since it was impossible to move the signs much closer to the orbitals than they are, we added dividing lines between the different images to clarified which are the occupied NTOs (electron) and which are the virtual ones (hole).
Comment:
- Why authors choose the listed solvents should be further clarified.
Response:
As we mentioned in the main text, these are solvents with widely different polarities. They are aprotic and quite appropriate for organic electrochemistry, with wide redox windows. On the practical side, the solubility of the electrolyte we use in these solvents is high enough for the concertation ranges we study.
Comment:
- More comparison to previous work should be provided.
Response:
We have been trying to keep the introduction and the background concise and focused. Nevertheless, to address the reviewer’s comment, we added a paragraph (lines 185 - 191) with extra references about the photoprobe that we chose for this study:
“Along with similar naphthalimide derivatives, ANI-Ph demonstrate remarkable features that make their photophysical behavior highly sensitive to the solvating environment [41-44]. As a “push-pull” conjugate comprising an electron-donating amine and an electron-withdrawing imide attached to the naphthalene rings, ANI-Ph undergoes intramolecular CT in its excited state. As a result, the emissive excited state of ANI-Ph possesses a large dipole moment in, making its fluorescence pronouncedly solvatochromic.”
Comment:
- There are too many too old references, which is better to be deleted or replaced with recent articles to show the novelty of this work.
Response:
We agree with the reviewer that we tend to add the original references, which are old, for many of the concepts and methodologies we use. We, however, respectfully disagree about the need to replace them with new ones. Often, the new references do not depict all important details in the methodologies (LMO, for example). Thus, it may prove helpful for some of the readers to go over the original publications describing the derivations and the assumptions for the development of the models used in the analysis, which warrants leaving these “old” references where they are.
Comment:
- There are still some typos and grammar issues in the manuscript. Authors should carefully recheck the whole manuscript.
Response:
Thank you for pointing it out. We went over the manuscript and corrected those on lines 57, 64, 66, 69,71,72,74,85, 127,128,131, 142, 158, 159,168,173, 177, 180, 186,190,211, 237, 239, 245,251, 252, 269, 286, 300, 289, 310,314,325, and 355.